# Influenza Vaccination Hesitancy among Healthcare Workers in South Al Batinah Governorate in Oman: A Cross-Sectional Study

**DOI:** 10.3390/vaccines8040661

**Published:** 2020-11-06

**Authors:** Salah T. Al Awaidy, Zayid K. Al Mayahi, Malak Kaddoura, Ozayr Mahomed, Nathalie Lahoud, Abdinasir Abubakar, Hassan Zaraket

**Affiliations:** 1Office of Health Affairs, Ministry of Health, P.O. Box 393, PC 100 Muscat, Oman; 2Directorate of Health Services, South Batinah Governorate, Ministry of Health, Oman, P.O. Box 543, PC 329 Rustaq, Oman; almayahi96@hotmail.com; 3Faculty of Medicine, Center for Infectious Diseases Research, American University of Beirut, 1107 2020 Beirut, Lebanon; malakkaddoura94@gmail.com; 4Department of Experimental Pathology, Faculty of Medicine, Immunology & Microbiology, American University of Beirut, 1107 2020 Beirut, Lebanon; 5Department of Public Health Medicine, University of KwaZulu Natal, Durban PC 4051, South Africa; ozayr411@gmail.com; 6Faculty of Public Health, Lebanese University, Fanar, P.O. Box 6573/14 Badaro, Lebanon; nathalie.lahoud@hotmail.com; 7World Health Organization Regional Office for the Eastern Mediterranean, Cairo 11371, Egypt; abubakara@who.int

**Keywords:** hesitancy, influenza, vaccination, uptake, healthcare workers, Oman

## Abstract

Background: Seasonal influenza infections are a major cause of morbidity and mortality worldwide. Healthcare workers (HCWs) are an important target group for vaccination against influenza due to their increased risk of infection and their potential to transmit the infection to their patients, families and communities. The aim of this study was to assess the potential hesitancy and its associated factors towards influenza vaccination amongst HCWs in the South Al Batinah governorate in Oman. Methods: A cross-sectional survey of 390 HCWs with direct or indirect patient contact was conducted in May and June 2019 using a self-administered questionnaire. Associations between HCW characteristics and vaccination status were examined using bivariate and multivariate analyses to identify the likelihood of vaccination against seasonal influenza among HCWs. Results: Overall, 60% of HCWs were vaccinated in the 2018/2019 season; vaccine uptake among nurses was 52% and uptake was higher among women. Self-protection and protection of the community were the most cited reasons for vaccine acceptance, with side effects being the main reason for hesitancy. Vaccinated respondents had a higher mean knowledge score (7.18; standard deviation SD: 2.14) than unvaccinated respondents (6.30; SD: 2.2). Odds of vaccination were highest among respondents who believed influenza vaccine should be mandatory for HCWs (Odds ratio (OR): 2.04 [1.30–3.18]), those working in the general medicine, emergency medicine, or intensive care units (OR: 1.92 [1.20–3.10]), nurses and doctors (OR: 1.75 [1.09–2.79]) and those who believe that HCWs should receive an influenza vaccine (OR: 1.35 [1.07–2.77]). Conclusions: The study provides valuable insights into the enablers and barriers of influenza vaccination practices among HCWs and may inform interventions to increase acceptance of vaccination.

## 1. Introduction

Seasonal influenza is an acute viral infection affecting all age groups worldwide and comprises a major disease burden in terms of morbidity, hospitalizations and deaths year-around. Persons aged ≥ 65, children aged ≤ 5, pregnant women and people with underlying chronic conditions are at high risk of severe disease and complications due to influenza [1].

The World Health Organization (WHO) estimates that influenza viruses infect up to one billion persons annually, causing an estimated 3–5 million severe cases [1]. Seasonal influenza is estimated to cause up to 650,000 respiratory deaths globally each year [2].

Annual seasonal epidemics have also been associated with substantial economic burden due to healthcare costs and productivity losses [3,4]. In Oman, it was estimated that influenza resulted in 3253 hospitalizations and 142 deaths in 2015. Influenza caused 27.5 (95% CI: 19.9–21.3) hospitalizations and 1.2 (95% CI: 0.9–1.5) deaths per 100,000 population. The incidence of influenza-associated hospitalization and death were highest among persons aged ≥ 65 years at 62.2 (95% CI: 53.2–71.1) and 11.3 (95% CI: 7.5–15.1) per 100,000 population, respectively, in 2015 [5].

Healthcare workers (HCWs) may have an increased occupational risk of influenza infection compared with the general population [6,7,8]. Infected HCWs may cause nosocomial outbreaks of influenza, leading to complications and death in high-risk patients [9,10,11]. Influenza infection among HCWs may also lead to absenteeism and disruption of medical services [12]. According to a report by the International Nursing Association, 7% of the all COVID-19 cases recorded worldwide are among HCWs [13]; this is equivalent to over 900,000 by 14 July 2020. These figures emphasize the high risk of infection among HCWs, particularly when vaccines to control an outbreak are not available. Establishing universal seasonal influenza vaccination programs among HCWs contributes to influenza pandemic preparedness by facilitating vaccine distribution and implementation mechanisms necessary to efficiently and quickly administer vaccines to this group and maintain an able population of front-line HCWs during pandemics [13]. Moreover, vaccinated HCWs are more likely to recommend the vaccine to their patients, which is critical for vaccine deployment during a pandemic [14]. 

In their 2012 recommendations, the Strategic Advisory Group of Experts (SAGE) on Immunization included HCWs among their priority groups for influenza immunization. SAGE also suggested that immunization of HCWs should be considered as part of a broader infection control strategy in healthcare facilities [15]. Similarly, the World Health Organization (WHO) prioritizes vaccination of HCWs against seasonal influenza to protect vulnerable patients and HCWs and to maintain the continuity of healthcare services [16]. 

Despite the health benefits of immunizing HCWs against influenza and the impact of their vaccine acceptance on that of their patients [17], misconceptions among HCWs toward seasonal influenza vaccines exist [14,18,19,20,21,22,23]. The Ministry of Health (MOH) of Oman established a no-cost national influenza vaccination program for HCWs in the public sector in 2011; the influenza vaccine is also provided free of charge to HCWs in the private sector. We sought to update our understanding of influenza vaccination coverage and attitudes among HCWs in Oman since the last such study was conducted in 2010, prior to the implementation of the national influenza vaccination program for HCWs [22]. These data are important to tailor local strategies to increase vaccine uptake in HCWs.

## 2. Methodology

**2.1. Study setting and population:** Oman is a small country with a total population of approximately 5 million and the total number of healthcare workers (HCWs) across the country is approximately 5500. We conducted a cross-sectional study among HCWs with direct or indirect patient contact in the public sector during the period of May to June 2019 in the South Al Batinah governorate. The governorate is estimated at a population of 437,818 and divided into six wilayats (districts) and contains 25 public sector health facilities (Figure 1). 

A sample size of 384 respondents was calculated using an estimated vaccine coverage of 50% with 80% power, with a 95% confidence interval (CI); we added an extra 5% to cater for non- or incomplete responses so our sample size increased to 400. We created three strata using probability proportional to size (PPS) to ensure representativeness at all levels of health facility—the regional referral hospital, three major polyclinics and all remaining health facilities. The proportion of each stratum was approximately; 30%, 30% and 40%, respectively. Questionnaires were distributed to a convenience sample of HCWs after a briefing about the study aims and replying to participants’ queries.

**2.2 Data collection:** A self-administered anonymous questionnaire was used to determine the knowledge and key factors underlying the HCWs’ practices related to acceptance of vaccination of oneself and to recommendations to patients. The questionnaire consisted of 28 open- and closed-ended questions covering demographics, knowledge about influenza disease and vaccines and, finally, attitudes toward influenza vaccines. Open-ended questions focused on reasons for vaccine refusal, willingness to receive vaccine, perceived barriers about influenza immunization, willingness to and confidence in recommending the vaccine to their patients at the institution. 

**2.3 Statistical analysis:** Statistical analyses were performed using SPSS version 21.0. We conducted descriptive analyses of social, demographic and other variables. Associations between HCWs’ social and demographic characteristics and vaccination status were examined using bivariate and multivariate analyses to identify predictors of vaccination against seasonal influenza among HCWs. We applied Pearson’s chi-square and Fisher’s exact tests to explore bivariate associations. We performed multivariate logistic regression to determine the adjusted odds ratios (aORs) and their 95% CIs. The fitness of the model was checked using omnibus and Hosmer and Lemeshow tests. 

The questionnaire included general statements about influenza and seasonal influenza vaccines using a five-point Likert scale (strongly agree to I don’t know). Based on these responses, we calculated a knowledge index score for HCWs. Scores were calculated based on correct (one point) and incorrect (zero points) responses to 11 statements, including influenza disease severity, transmission, benefits of vaccination and recommendations. The median index score and the interquartile range (IQR) were reported. Unadjusted and adjusted odds ratios were calculated to assess the relationship of knowledge with the willingness to receive and recommend the influenza vaccine to patients. 

**2.4 Ethical approval:** Ethical clearance was obtained from the South Al Batinah governorate ethical committee and from the Institutional Review Board at the American University of Beirut. Informed consent was obtained for every participant. The participants were reassured of the confidentiality of the collected information, and involvement in the study was voluntary.

## 3. Results

In total, 390 questionnaires were completed and returned. Women accounted for 77.9% (304) of respondents; 86 questionnaires (22.1%) were completed by men. Most participants were between 25 and 39 years old (317 participants; 81.3%). Among the participants, 9.5% had a master’s degree and 1.8% were doctorate holders (Table 1). 

Nurses accounted for the greatest number of respondents (173 participants; 44.4%), followed by doctors, residents and interns (108 participants; 27.7%). Thirty-seven (9.5%) participants were pharmacists and 56 (14.4%) were laboratory and X-ray technicians.

The majority (*n* = 192; 49.2%) of respondents did not belong to one of the specified departments, 103 (26.4%) participants worked in general internal medicine, 22 (5.6%) in radiology and 18 (4.6%) in the emergency department. Eighteen (4.6%) were specialists in obstetrics/gynecology and 16 were pediatric specialists (4.1%). Almost one third of the participants (35.6%) reported seeing more than 40 patients per day. Moreover, most participants had at least 5 years of experience in healthcare; 139 (35.6%) and 99 (25.4%) participants had 5 to 9 and 10 to 14 years of experience, respectively (Table 1).

### 3.1. Influenza Vaccination Status

Of the 390 responses that we received, information on influenza vaccination status was available for 353 participants and were included in the subsequent analysis. Sixty percent (*n* = 213) of the study population with complete information was vaccinated in the 2018/2019 season (Figure 2, Table 1). Nurses constituted the highest proportion of vaccinated HCWs (110 participants; 52%), followed by doctors, residents and interns (60 participants; 28.4%) other healthcare personnel (Table 1). Employees from the general medicine department, emergency medicine and intensive care units were significantly more likely to be vaccinated against influenza (*p* = 0.004). 

There was significantly more uptake of vaccine among female HCWs compared to males (*p* = 0.021); paramedical staff, including laboratory and X-ray technicians, had a lower uptake of influenza vaccine compared to doctors and nurses (*p* = 0.016); HCWs with a patient volume of 11–20 or 21–30 had significantly more vaccination uptake (*p* = 0.013) than those with <10 or >40 patients. Age, hospital department, years of work, years since graduation from university and level of education did not show any statistically significant effect on the uptake of the vaccine (Table 1).

### 3.2. Willingness to Be Vaccinated and/or Recommend Vaccination to Patients

Three hundred and twenty respondents in the public sector of the South Al Batinah governorate (83%) reported willingness to be vaccinated against influenza. Three hundred and seventeen (81%) of the HCWs were willing to recommend the vaccine to their patients (Figure 2). 

The barriers and enablers of vaccine acceptance and recommendation are summarized in Figure 3. The three main reasons for willingness to be vaccinated were reported by the enrolled HCWs: the need to protect themselves, those around them and their family (59.0%), high risk of exposure to infection (17.0%) and availability of vaccine (6.7%). The most frequently cited causes of hesitancy included fear of side effects (33.6%), fear of pain (12.6%), concerns regarding vaccine efficacy due to viral evolution (6.1%) and lack of knowledge about influenza (6.1%).

The main reasons cited for recommending the influenza vaccine to patients included protection of the patient, surrounding people and their family (50.4%), reducing illness and improving health (15.8%) and protection of the elderly and patients with chronic diseases against viral respiratory tract infections (11.3%). Among HCWs who do not recommend the influenza vaccine to patients, the most cited reasons were potential side effects (35.2%), fear of allergic reaction or pain (9%) and perceived lack of benefit (8.1%) (Figure 4). 

### 3.3. Knowledge about Influenza Vaccine

The mean knowledge score for participants was 6.72/11 (SD: 2.3), with a median score of 7 (IQR: 6–8) (Table 1). Among the respondents, 59.5% had a knowledge score greater than the mean. Vaccinated HCWs had a significantly (*p* < 0.01) higher mean knowledge score (7.18; SD: 2.14) than those who were unvaccinated (6.30; SD: 2.2) during than the 2018–2019 season. Thirty-three percent of respondents (*n* = 130) did not believe influenza could result in serious illness, while 42% (*n* = 163) believed that influenza does not cause much illness among HCWs in Oman. While influenza is spread through respiratory droplets, 39% (*n* = 153) of the respondents believed influenza could spread through body fluids. Thirty percent (*n* = 118) of the respondents were of the opinion that providing the influenza vaccine to HCWs would not reduce absenteeism, while 37% (*n* = 146) indicated that the influenza vaccine for HCWs will prevent influenza-infected patients from developing severe illness or dying from the disease. A further 28% (*n* = 110) of respondents indicated that the influenza vaccine can cause a person to become ill with influenza (Figure 5). 

Regarding attitudes towards influenza vaccination, 72% (*n* = 284) of the respondents indicated that HCWs should receive the influenza vaccine, with 63% (*n* = 247) indicating that the influenza vaccine should be mandatory. 

### 3.4. Association between Socio-Demographic Factors, Knowledge Scores, Attitudes and Influenza Vaccine Uptake in the 2018–2019 Season

In the bivariate analysis, nurses and doctors were grouped together and compared to other HCWs. Nurses and doctors (OR: 1.75 [1.09–2.79]), HCWs from the general medicine, emergency medicine, intensive care units (OR: 1.92 [1.20–3.10]), participants who believed that HCW’s should receive influenza vaccine (OR: 1.35 [1.07–2.77]) and participants who believed that influenza vaccine should be mandatory for HCWs (OR: 2.04 [1.30–3.18]) were significantly more likely to be vaccinated for influenza in the 2018–2019 season. Participants with a university or postgraduate degree were less likely (OR: 0.79 [0.48–1.30]) to be willing to receive vaccine compared to participants with technical or undergraduate degrees only, though this finding was not statistically significant. Although also not statistically significant, HCWs with knowledge scores > 7 were more likely to be vaccinated. After controlling for confounding and interaction, HCWs from the general medicine, emergency medicine and intensive care units (OR: 1.71 [1.05–2.78]) and participants who believed that influenza vaccine should be mandatory (OR: 1.90 [1.17–3.10]) were significantly more likely to be vaccinated for influenza in the 2018–2019 season (Table 2).

In the bivariate analysis, nurses and doctors (OR: 2.49 [1.39–4.42]), participants vaccinated in the 2018/2019 season (OR: 3.27 [1.84–5.88]), those with knowledge scores >7 (OR: 2.28 [1.26–4.25]), participants who believed that HCWs should receive influenza vaccine (OR: 8.37 [4.54–15.56]) and participants who believed that influenza vaccine should be mandatory (OR: 4.47 [2,49–8.16]) were significantly more likely to be willing to receive the vaccine if provided for free, while participants with a postgraduate degree were less likely (OR: 0.45 [0.20–0.94]) to be willing to receive the vaccine. After controlling for confounding and interaction, females (OR: 0.44 [0.19–0.99]) were significantly less likely to be willing to receive the vaccine, while participants vaccinated in the past season (OR: 1.78 [1.17–2.76]), participants who believed that HCWs should receive the influenza vaccine (OR: 5.5 [2.73–11.09]) and participants who believed that influenza vaccine should be mandatory (OR: 2.02 [1.02–3.98]) were significantly more likely to be willing to receive the vaccine if it was provided for free (Table 2).

### 3.5. The Association between Socio-Demographic Factors, Knowledge Scores and Attitudes with the Willingness to Recommend Influenza Vaccination in Patients

Participants who believed that HCWs should receive influenza vaccine (OR: 7.90 [4.38–14.36]), those who were willing to receive the vaccine (OR: 7.26 [3.88–14.39]) and nurses and doctors (OR: 3.32 [1.88–5.80]) were more likely to recommend vaccination to patients, while participants with a university or postgraduate degree were less likely (OR: 0.31 [0.12–0.68]) to recommend vaccination to their patients. After controlling for confounding and interaction, HCWs with >10 years of service (OR: 2.43 [1.39–4.31]), participants who believed that HCWs should receive influenza vaccine (OR: 4.69 [2.20–9.99]) and participants who were willing to receive the vaccine (OR: 3.25 [1.61–6.59]) were significantly more likely to recommend vaccination to patients (Table 2).

## 4. Discussion

The universal influenza vaccine program in Oman strives for 99% coverage of all HCWs. In our study, only 60% of the HCWs surveyed in the South Al Batinah governorate were vaccinated for the 2018–2019 season, despite the availability of free vaccine to all HCWs at their workplaces. While vaccine coverage remains sub-optimal in Oman, it has improved since a 2009 estimate (46.4%) by Abou Gharbieh et al. [22] and is higher than rates observed in some neighboring countries with free HCW vaccination programs, including an estimated 53.4% coverage reported in the Dubai Health Authority (UAE) for the 2016–2017 season [24]. In Iran, a recent survey reported a vaccination coverage of 57.7% in the capital, Tehran, in the 2015–2016 season [25]. In Qatar, a vaccination campaign was able to achieve 77% coverage among HCWs during the 2015–2016 season [26]. Therefore, free-of-charge vaccination alone is not enough to attain optimal vaccination coverage among HCWs. 

Consistent with our findings, a systematic literature review found that males were more likely to intend to receive vaccine [27]; however, this did not correlate with higher vaccination uptake in our study. HCWs with >10 years of service were significantly more likely to recommend influenza vaccination to their patients; however, no association was found between years of service and vaccine uptake or willingness to receive vaccine. Similar findings were observed in a hospital in Singapore, where length of service did not correlate with greater compliance with vaccination [28]. 

Almarzooki et al. reported an association between professional occupation of HCWs and vaccine uptake in the UAE, with physicians having the highest uptake, followed by nurses [24]. We also found that physicians and nurses as well as HCWs across all categories working in general medicine, emergency medicine or intensive care units were more likely or willing than others to be vaccinated and to recommend the influenza vaccine to patients. A number of studies have corroborated the association between being a medical doctor and influenza vaccine uptake, willingness to recommend or to be vaccinated themselves [29,30,31]; however, the literature has shown contrary findings to our study with respect to the association between being a nurse and vaccine uptake [31]. Nonetheless, a survey at a hospital in Singapore indicated that the nursing staff were significantly more likely to be vaccinated [28]. 

The main reasons cited for the HCW uptake or willingness to be vaccinated include protection of themselves, surrounding people or family, high risk of exposure to infection, vaccine availability, reduction in illness and improved health, protection of the elderly and patients with chronic diseases against viral respiratory tract infections and presence of co-morbidities. The main causes of hesitancy in influenza vaccine uptake, willingness to be vaccinated and/or recommending the vaccine to patients included side effects, perceived lack of benefit and the risk of vaccinated patients developing vaccine-induced influenza illness. The factors influencing influenza vaccine uptake and recommendation as well as the factors linked to hesitancy for uptake or willingness to recommend to patients are not unique to Oman. Similar enablers and barriers to vaccination were cited by HCWs in Dubai Health Authority (UAE) [23] and Saudi Arabia [32]. A systematic review and meta-analysis conducted to determine the prevalence of influenza vaccination among nurses and ancillary workers in Italy produced similar findings [33]. 

In this study, knowledge was a key factor that influenced HCWs’ willingness to be vaccinated as well as recommend the influenza vaccine to patients. These findings were consistent with the findings of a study in Dubai (UAE) that showed a higher proportion of HCWs with good knowledge accepted the vaccine, while poorer levels of knowledge resulted in lower acceptance of the vaccine. HCWs with good levels of knowledge recommended the influenza vaccine to their patients [23]. However, we did not find a significant correlation between knowledge score and vaccine uptake during the 2018–2019 season. Deficiencies in HCWs’ general knowledge of influenza were noted in the study, particularly beliefs regarding transmission of influenza through body fluids and the potential for influenza to cause severe illness or death. These gaps in knowledge underscore the importance of the education of HCWs regarding influenza. 

Seventy-two percent of the HCWs in our study were in favor of mandatory vaccination of HCW. Similarly, in neighboring Saudi Arabia, the majority (83%) of surveyed HCWs were supportive of universal and mandatory influenza vaccination [34]. A positive attitude towards influenza vaccination by HCWs was significantly associated with the willingness to be immunized against influenza and recommending influenza vaccination to patients. A similar finding of a significant relationship between a positive sentiment toward universal vaccination of HCWs and influenza vaccine uptake was shown in a survey of HCWs in Ireland [35]. A multi-center cross-sectional study in Saudi Arabia showed that recommendations from the government on influenza vaccination for HCWs and mandatory requirements were important predictors of vaccine acceptance [36].

A positive attitude from HCWs towards vaccination has been linked to improved vaccine acceptance [14]. Consistently, we found that Omani HCWs who were willing to receive the vaccine were also more likely to recommend it to their patients. Therefore, promoting vaccine acceptance and uptake among HCWs should be a core component of pandemic preparedness both to protect them and to promote vaccination among the general population during a pandemic [37]. 

It should be noted that our study is limited by its reliance on self-reporting by HCWs, which could be associated with recall bias and potentially influenced by social desirability. Furthermore, our study did not capture HCWs from the private sector. However, South Al Batinah has a very small private sector and, therefore, our data are representative of the majority of HCWs in the region. The small sample size has an effect on the power of the study as well as the significance of the results, thereby limiting the generalizability of the conclusions. The use of the mean influenza knowledge score as an indicator for vaccine practices might not accurately represent the general level of knowledge. 

## 5. Conclusions and the Way Forward

To the best of our knowledge, this is the first study to assess the knowledge, attitudes and practices of HCWs regarding the seasonal influenza vaccine in Oman. Vaccination coverage among HCWs was suboptimal, despite its accessibility and availability free of charge. It is crucial to improve HCWs’ personal confidence and knowledge in vaccination and engage them in activities targeting vaccine hesitancy among their patients. The barriers and misconceptions about seasonal influenza should be tackled through low-cost and accessible educational interventions to increase seasonal influenza vaccination uptake at the individual and societal levels. The use of newsfeed and reminders as a nudge to remind HCWs of vaccination to be done in advance of the influenza season or in the early part of the season could also enhance uptake. Furthermore, non-financial incentives and timely feedback on the vaccination uptake levels by professionals at their workplace and reasons for poor uptake can help advocate for the vaccine. 

## Figures and Tables

**Figure 1 vaccines-08-00661-f001:**
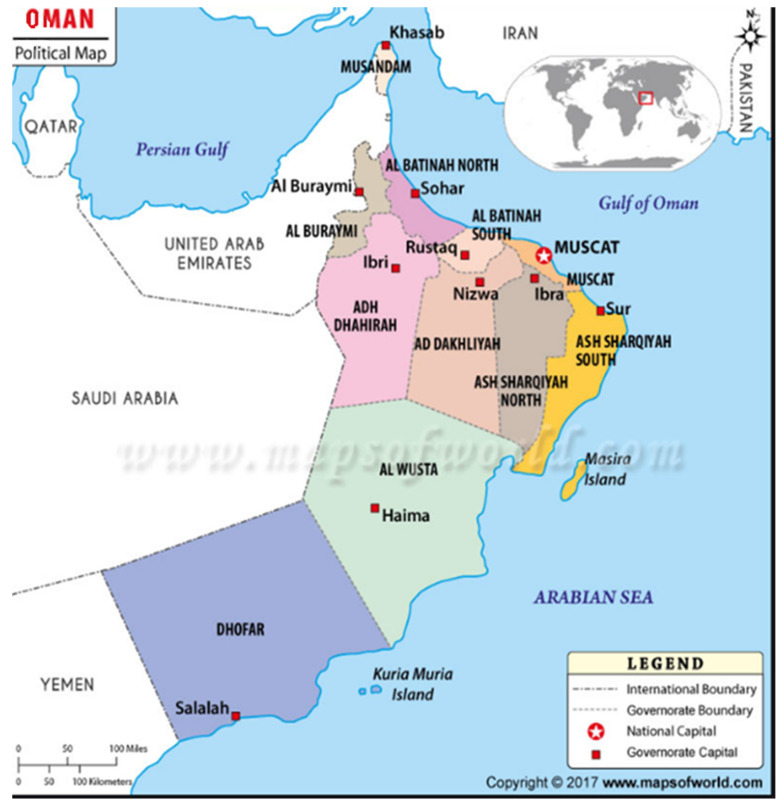
Map of Oman.

**Figure 2 vaccines-08-00661-f002:**
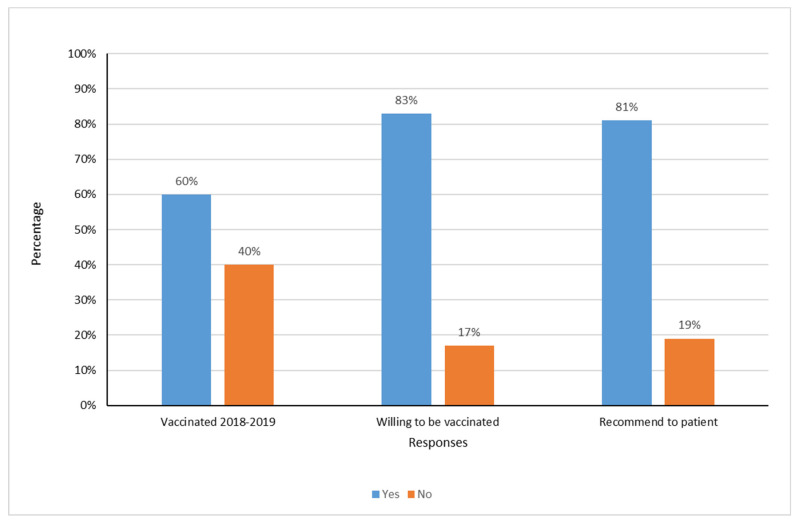
The proportion of HCWs vaccinated, willing to vaccinate and/or recommend the influenza vaccine to patients.

**Figure 3 vaccines-08-00661-f003:**
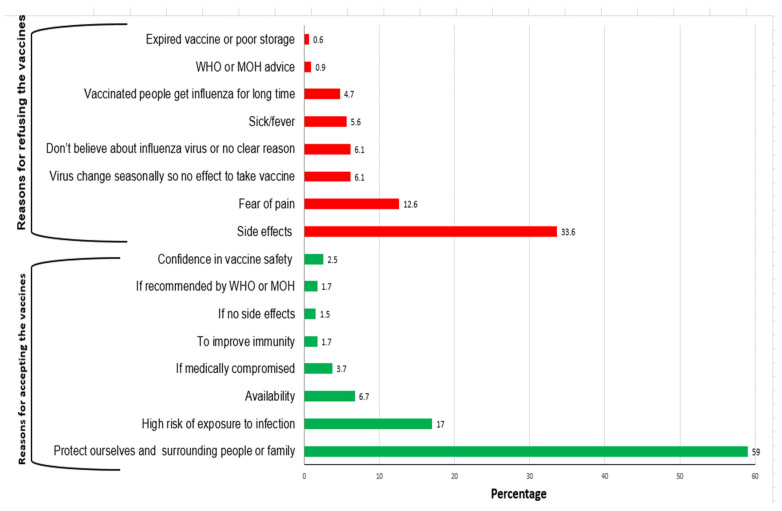
The reasons for accepting or refusing the vaccine among health care workers.

**Figure 4 vaccines-08-00661-f004:**
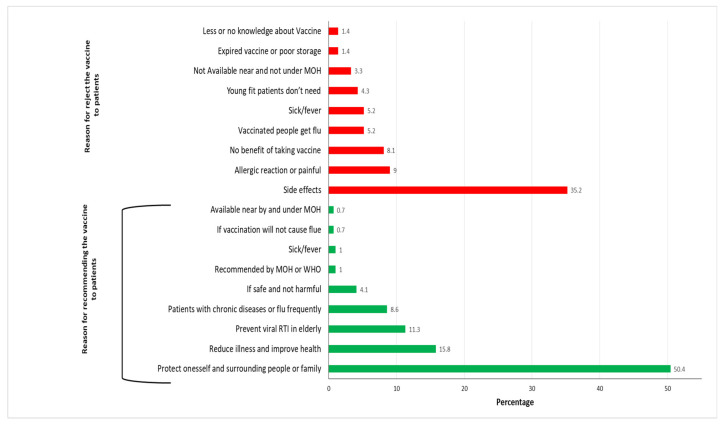
The reasons for recommending and rejection of the vaccine to patients.

**Figure 5 vaccines-08-00661-f005:**
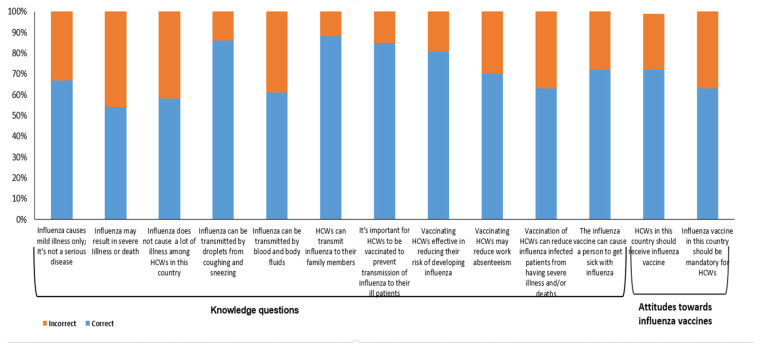
Frequency of knowledge responses and attitudes towards influenza vaccine amongst healthcare workers (HCWs) for the 2018–2019 season, *n* = 390.

**Table 1 vaccines-08-00661-t001:** Frequency table of demographic characteristics of the health care workers categorized by vaccination uptake.

Characteristics	Study Population *n* = 390 *		Vaccine Taken *n* = 213 (54.6%)		Vaccine Not Taken *n* = 140 (35.9%)		χ^2^	*p*
Age (years)								
	*n*	%	*n*	%	*n*	%		
18–29	9	2.3	1	11.1	7	77.7	11.1	0.05
25–29	89	22.8	42	47.1	30	33.7		
30–34	131	33.6	72	55.0	49	37.4		
35–39	97	24.9	57	58.7	34	35.0		
40–49	52	13.3	35	67.3	14	26.9		
50–59	12	3.1	6	50.0	6	50.0		
Gender								
Male	86	22.1	36	41.9	38	44.2	5.4	0.021
Female	304	77.9	177	58.2	102	33.6		
Occupation								
Nurse	173	44.4	110	63.6	53	30.6	13.9	0.016
Doctor/resident/intern	108	27.7	60	55.6	39	36.1		
Pharmacist	37	9.5	18	48.6	16	43.2		
Lab technician	33	8.5	11	33.3	14	42.4		
X-ray technician	23	5.9	5	21.7	12	52.2		
Others	16	4.1	7	43.7	4	25.0		
Working department								
General/internal medicine	103	26.4	67	65.0	30	29.1	27.81	0.033
Emergency department	18	4.6	9	50.0	6	33.3		
Obstetrics/gynecology	18	4.6	11	61.1	7	38.8		
Adult intensive care unit	7	1.8	6	85.7	0	0		
Neonatal intensive care unit	2	0.5	1	50.0	1	50.0		
Pediatrics	16	4.1	8	50.0	8	50.0		
Radiology	22	5.6	5	22.7	4	18.2		
Surgery	9	2.3	5	55.6	4	44.5		
Other	192	49.2	98	51.0	73	38.0		
Did not answer	3	0.8		0		0		
Volume of patients								
<=10	49	12.6	21	42.9	23	46.9	12.7	0.013
11–20	62	15.9	41	66.1	15	24.2		
21–30	73	18.7	52	71.2	20	27.4		
31–40	54	13.8	28	51.9	20	37.0		
>40	139	35.6	68	48.9	56	40.3		
Did not answer	13	3.3	-	-	-	-		
Highest education level								
College or university	247	63.3	133	53.8	89	36.0	0.46	0.927
Master’s degree	37	9.5	21	56.8	15	40.6		
Doctorate	7	1.8	4	57.1	3	42.9		
Other	95	24.4	54	56.8	31	32.6		
Did not answer	4	1.0	-	-	-	-		

* 39 healthcare workers (HCWs) did not report their vaccination history.

**Table 2 vaccines-08-00661-t002:** The association between socio-demographic factors, knowledge scores and attitudes and influenza vaccine related practices.

	Influenza Vaccine Uptake						Willingness to Vaccinate						Recommend to Patients					
Characteristics	Unadjusted Odds Ratio	95% CI	*p* Value	Adjusted Odds Ratio	95% CI	*p* Value	Unadjusted Odds Ratio	95% CI	*p* Value	Adjusted Odds Ratio	95% CI	*p* Value	Unadjusted Odds Ratio	95% CI	*p* Value	Adjusted Odds Ratio	95% CI	*p* Value
Females * versus males	1.54	0.91–2.57	0.08	1.53	0.93–2.53	0.09	0.56	0.24–1.17	0.11	0.44	0.19–0.99	0.046	0.99	0.50–1.88	0.97			
Age > 30 years * versus < 30 years	1.27	0.78–2.08	0.3	1.04	0.63–1.71	0.89	1.54	0.83–2.81	0.13	1.31	0.67–2.58	0.44	2.03	1.13–3.61	<0.001	0.88	0.43–1.83	0.74
Nurses and doctors * versus others	1.75	1.09–2.79	0.01	1.32	0.80–2.18	0.3	2.49	1.39–4.42	<0.001	1.78	0.90–3.54	0.1	3.32	1.88–5.80	<0.001	1.9	0.96–3.76	0.07
University and above education * versus and below	0.79	0.48–1.30	0.33	0.78	0.47–2.19	0.33	0.45	0.20–0.94	0.03	0.52	0.23–1.17	0.12	0.31	0.12–0.68	<0.001	0.43	0.18–1.03	0.06
General medicine, emergency medicine and intensive care * versus others	1.92	1.20–3.10	0.004	1.71	1.05–2.78	0.03	1.75	0.94–3.45	0.07	1.15	0.62–2.57	0.7	2.66	1.36–5.51	0.002	1.91	0.88–4.14	0.1
>30 patients daily versus < 30 patients	0.75	0.42–1.31	0.27				0.75	0.42–1.31	0.27				0.97	0.56–1.67	0.91			
>10 years of service * versus < 10 years of service	0.83	0.54–1.28	0.38				1.25	0.72–2.20	0.39				2.43	1.39–4.31	<0.001	2.79	1.39–5.59	0.004
Vaccinated in 2018–2019 * versus not vaccinated							3.27	1.84–5.88	<0.001	1.78	1.17–2.76	0.008	1.88	1.09–3.25	0.02	1.03	0.68–1.54	0.89
Knowledge score > 7 * versus < 7	1.35	0.88–2.08	0.15	1.1	0.71–1.72	0.68	2.28	1.26–4.25	0.004	1.12	0.57–2.20	0.75	2.4	1.34–4.39	0.002	1.14	0.57–2.24	0.72
HCWs should receive influenza vaccine (yes * versus no)	1.72	1.07–2.77	0.02	1.06	0.61–1.85	0.82	8.37	4.54–15.56	<0.001	5.5	2.73–11.09	<0.001	7.9	4.38–14.36	<0.001	4.69	2.20–9.99	<0.001
Influenza vaccine should be mandatory (yes * versus no)	2.04	1.30–3.18	<0.001	1.9	1.17–3.10	0.01	4.47	2.49–8.16	<0.001	2.02	1.02–3.98	0.04	3.11	1.78–5.44	<0.001	1.17	0.56–2.43	0.68
Willingness to be vaccinated													7.26	3.88–13.49	<0.001	3.25	1.61–6.59	<0.001

* Indicates the reference group.

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
