# Peer review of "Influenza Vaccination Hesitancy among Healthcare Workers in South Al Batinah Governorate in Oman: A Cross-Sectional Study"

_vaccines, 2020, doi:10.3390/vaccines8040661_

Round 1

Reviewer 1 Report

The reviewer thanks the authors for their responses to manuscript comments.  

It would be beneficial to add Oman total as well as HCW population statisitics to the manuscript.  The estimated proportion of the HCW population that participated is useful information for the reader. 

Author Response

Highly appreciated the reviewer's comments. As requested, we added the total population in Oman as well as total health care workers. See page 7

Reviewer 2 Report

Dear authors

About the tables:

Table 1

Frist row:  Total   390  100%      213  54.6%     140   45.4%.  Percentages are missing.

In addition, The second column misses the symbol %,

Use any decimal or one for describing percentages. Please standardize the criteria.

Table 2

Two groups are compared, but the reference group is not known. For example, Males vs. Females, but does an OR of 1.54 mean more or less risk of being vaccinated for males? Simply putting an asterisk, or other indication, on the reference group would be sufficient.

Kind regards

Author Response

We thank the reviewers for their observations.

Table 1, we revised the value for consistency and do the modification as per the advice

Table 2, we indicated the reference group with an asterisk

The changes made we highlighted in YELLOW

This manuscript is a resubmission of an earlier submission. The following is a list of the peer review reports and author responses from that submission.

Round 1

Reviewer 1 Report

In the manuscript by Awaidy et al.  the uptake of IAV vaccines was surveyed among health care workers in Oman.  The study is of importance to public health and evaluates the reasons why individuals decide not to get vaccinated against IAV. 

The study is limited in n for a survey based study (n for powerful data is estimated within to be a little under 400) in comparison to recent publication in Vaccines evaluating similar parameters in China where the n's were 10X higher. (Xu et al February of 2020).

The n of surveys including vaccination information fell below the power cut-off.  A number of the conclusions made within the manuscript were stated as not reaching significance.  Greater n will support more confidence in the statements and conclusions of the authors. 

Another season of data to repeat the findings and build n would be beneficial. 

Reviewer 2 Report

Dear authors

I have read the manuscript entitle “Influenza Vaccination Hesitancy among Health Care Workers in South Al Batinah governorate in Oman: A cross-sectional study” and I have some recommendations and a couple of doubts.

About the tables and figures:

Table 1 should include a row with the figures of the total. I think this should be the first row.

Figures 3, 4 and 5 should be changed by tables; the size-print of the figures is too small to be readable.

Table 2, in the logistic regression applied to calculate ORs, which are the reference groups? In any case, these reference groups should be mentioned in the table.

About the Knowledge Index Score: Has been this scale validated? Is it based in others studies? I have some doubts about this score could measure the Knowledge. At least, this fact should be mentioned at limitations paragraph (line 302)

Kind regards